# AlphaD3M: An Open-Source AutoML Library for Multiple ML Tasks

Roque Lopez[1]  Raoni Lourenço[2]  Remi Rampin[1]  Sonia Castelo[1]  Aécio Santos[1]  Jorge Ono[1]
Claudio Silva[1]  Juliana Freire[1]

[1]New York University
[2]University of Luxembourg

**Abstract**  We present AlphaD3M, an open-source Python library that supports a wide range of machine learning tasks over different data types. We discuss the challenges involved in supporting multiple tasks and how AlphaD3M addresses them by combining deep reinforcement learning and meta-learning to construct pipelines over a large collection of primitives effectively. To better integrate the use of AutoML within the data science lifecycle, we have built an ecosystem of tools around AlphaD3M that support user-in-the-loop tasks, including selecting suitable pipelines and developing custom solutions for complex problems. We present use cases that demonstrate some of these features. We report the results of a detailed experimental evaluation showing that AlphaD3M is effective and derives high-quality pipelines for a diverse set of problems with performance comparable or superior to state-of-the-art AutoML systems.

## 1 Introduction

Automated Machine Learning (AutoML) has emerged as an alternative to automatically synthesize machine learning (ML) pipelines, thereby democratizing ML techniques to non-experts as well as increasing the productivity of data scientists. Different approaches have been proposed for AutoML systems. Some focus on specific components of an ML pipeline, such as hyperparameter optimization or model selection, while others, given a dataset and a prediction task, generate end-to-end pipelines that encompass data pre-processing, feature, and model selection (Hutter et al., 2019). Most end-to-end systems are designed to work with tabular data and only support classification and regression problems (Feurer et al., 2015; LeDell and Poirier, 2020; Olson and Moore, 2016; Kotthoff et al., 2017). Cloud AutoML (Google Cloud AutoML, 2020) and AutoGluon (Erickson et al., 2020) also create pipelines to classify text and images and perform object detection tasks. However, these systems do not support more complex data types such as graphs, time series, audio, and video, limiting the types of problems they can address. Table 1 shows the set of task types supported by different AutoML systems.

In the context of DARPA's Data-Driven Discovery of Models (D3M) program (Elliott, 2020), several AutoML systems have been developed to support a wide range of data types and ML tasks using an extensive set of computational primitives as building blocks – we refer to these as *multi-task* AutoML systems (MT-AutoML). MT-AutoML systems face an essential challenge: effectively searching an ample space of primitives required to synthesize pipelines for a broad range of tasks and data types. To prune the search space, many D3M MT-AutoML systems use manually-crafted templates and grammars (D3M, 2022) that prescribe combinations of primitives that make sense for different problems. This, in turn, leads to other challenges: creating these templates or grammars is not only time-consuming but failing to include the necessary rules that cover the relevant primitives (and their combination) for multiple task types can negatively impact the ability of an MT-AutoML system to derive performant pipelines.

Table 1: Tasks supported by different AutoML Systems.

| Systems | Tabular Classification | Text classification | Image classification | Audio classification | Video classification | Tabular Regression | Clustering | Time series forecasting | Time series classification | Object detection | LUPI | Community detection | Link prediction | Graph matching | Vertex classification | Collaborative filtering | Semisupervised classification |
|---|---|---|---|---|---|---|---|---|---|---|---|---|---|---|---|---|---|
| AutoGluon | ✓ | ✓ | ✓ | | | ✓ | | ✓ | | ✓ | | | | | | | |
| AutoWEKA | ✓ | | | | | ✓ | | | | | | | | | | | |
| Auto-Sklearn | ✓ | | | | | ✓ | | | | | | | | | | | |
| Cloud AutoML | ✓ | ✓ | ✓ | | ✓ | ✓ | | | | ✓ | | | | | | | |
| H2O | ✓ | ✓ | | | | ✓ | | | | | | | | | | | |
| TPOT | ✓ | | | | | ✓ | | | | | | | | | | | |
| **AlphaD3M** | ✓ | ✓ | ✓ | ✓ | ✓ | ✓ | ✓ | ✓ | ✓ | ✓ | ✓ | ✓ | ✓ | ✓ | ✓ | ✓ | ✓ |

We present AlphaD3M, an open-source AutoML library[1] that supports a wide range of data and problem types (see Table 1). AlphaD3M introduces new techniques to navigate the large search spaces MT-AutoML systems must navigate effectively. They include an algorithm that applies meta-learning to automatically derive task-based context-free grammars (CFGs) which cover a multitude of problems; and a novel search strategy that, based on previously generated pipelines and their performance, prioritizes primitives that are correlated with good pipeline performance.

AlphaD3M includes components that aim to support usability and integration with other tasks in the data science lifecycle, from data exploration and model summarization to model deployment. It is possible to extend AlphaD3M and combine it with other tools through its flexible API. For example, its integration with the PipelineProfile (Ono et al., 2021) allows users to explore and compare the set of derived pipelines visually. Besides describing the API and these components, we also present case studies demonstrating how users can improve the ML solutions via interaction in AlphaD3M.

We conducted a detailed experimental evaluation to assess the ability of AlphaD3M to handle a rich set of tasks and data types as well as to compare its performance against state-of-the-art AutoML and MT-AutoML systems. We used two benchmarks: (a) a collection of 112 datasets that covers seventeen different ML tasks, and (b) the OpenML AutoML Benchmark for tabular classification problems. Our results show that the search strategies used by AlphaD3M are effective: the system generates pipelines whose performance is superior or on par with those derived by other systems, including systems that focus on a small set of problems and have to navigate a much smaller search space.

## 2 Related Work

**Task Coverage**. Many AutoML systems have been proposed to work with tabular data, for example: Auto-sklearn (Feurer et al., 2015), TPOT (Olson and Moore, 2016), and H2O (LeDell and Poirier, 2020). The deep reinforcement learning algorithm proposed by Drori et al. (2019) aimed to support multiple learning tasks and data types, however, its implementation was limited to classification and regression tasks over tabular and text data. AutoML systems developed in industry, such as Cloud AutoML by Google and AutoGluon by Amazon, handle text and image data, but still support a limited number of learning tasks. In contrast, AlphaD3M supports a wide range of data types (tabular, text, images, audio, video, and graph) and a rich set of ML tasks as shown in Table 1.

**Data and Model Exploration**. Interactive data analytics systems such as Visus (Santos et al., 2019), TwoRavens (Gil et al., 2019), and Snowcat (Cashman et al., 2018) have been developed to guide users throughout the model-building process, from exploring the input data to comparing the ML pipelines produced by AutoML systems. They target primarily domain experts who have little or

---

[1] https://gitlab.com/ViDA-NYU/d3m/alphad3m

no expertise in ML and thus lack support for the customization of pipelines for complex problems. These systems trade off flexibility for ease of use. As such, they are limited to the operations implemented in their visual interfaces; extensive and time-consuming changes in their workflows are required to support new data types and tasks (e.g., graph data). Other approaches mimic the interface of traditional ML libraries, through which developers often build a single solution for a given task (Grafberger et al., 2021). AlphaD3M allows ML experts to explore the derived pipelines and customize them through a user-friendly interface within a Jupyter Notebook environment. In addition, instead of retrieving only the best pipeline, AlphaD3M returns all valid pipelines, ranks, and presents them to the user for comparison, refinement, and selection.

## 3 The AlphaD3M Library

AlphaD3M is a *multi-task* AutoML system. It is implemented in Python and can be used via *pip* installation or Docker. Figure 1 shows an overview of this library and its components. To build ML pipelines, AlphaD3M uses a rich set of primitives and a meta-learning database from the D3M ecosystem D3M

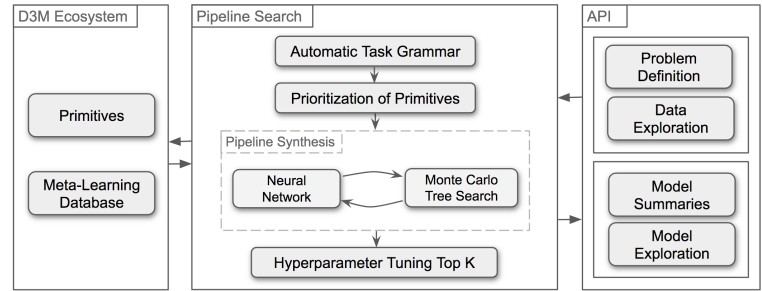

Figure 1: Overview of AlphaD3M.

(2022). The pipeline search is conducted by four modules which: (a) automatically construct of task-specific grammars; (b) prioritize primitives that are more likely to be effective; (c) synthesize pipelines using Monte Carlo Tree Search and Neural Networks (Drori et al., 2019); and (d) tune hyperparameters. The library implements a Python API through which users can define the problem to be solved, explore the input data, obtain model summaries, analyze and compare the produced pipelines, as well as improve and deploy them.

### 3.1 The D3M Ecosystem

**Primitives**. AlphaD3M uses a comprehensive collection of primitives developed by performers in the D3M program as well as from open-source libraries (e.g., scikit-learn). In total, there are 312 primitives available for different steps in ML pipelines, including data pre-processing, feature extraction, feature selection, prediction, and clustering (D3M Primitives, 2022), and implement state-of-the-art methods, such as ResNet50 (He et al., 2016), ARIMA (Wilson, 2016), among others. **The Marvin Meta-Learning Database**. Marvin is an open corpus of curated ML pipelines, datasets, and problems (Marvin, 2020). All pipelines in Marvin share the same set of primitives and are specified using the D3M format. Marvin stores approximately 2.5 million pipelines executed over 600 datasets. Since data scientists and AutoML systems that use different search strategies have produced these pipelines, the database covers a wide variety of pipeline patterns. As discussed below, we leverage the data in Marvin to assist in and improve the AlphaD3M search process. To the best of our knowledge, ours is the first work that explores this corpus.

### 3.2 Pipeline Search

The automatic synthesis of pipelines is a combinatorial problem in which we must find the best combinations of primitives and their hyperparameters. With 312 primitives and over 1,500 hyperparameters in the D3M ecosystem, the search space becomes prohibitively large. For instance, considering just the classification task over tabular data, there are 22 data cleaning, 87 data transformation, and 44 classifier primitives, leading to 84,216 possible pipelines to test. AlphaD3M uses a multi-pronged approach to manage this search space described below.

**A**  **Pipeline Synthesis Using Monte Carlo Tree Search and Neural Networks**. To synthesize the ML pipelines, AlphaD3M uses the strategy introduced by Drori et al. (2019), which is based on a single-player game technique inspired by AlphaZero (Silver et al., 2017). It applies model-based reinforcement learning with a neural network sequence model, and a Monte Carlo Tree Search (MCTS). The metadata encoding the pipeline, the dataset, and the task are analogous to an entire game board configuration in AlphaZero. The possible game states consist of all valid pipelines generated from a set of primitives and modified by actions guided by a manually-designed CFG. The model outputs a sequence of primitives. Pipelines are constructed by an LSTM. Given a state $s$ composed of a vector encoding the whole board configuration (dataset, task, pipeline), the neural network predicts the probabilities $P(s, a)$ over actions $a$ from a state $s$. This process produces a set of action sequences $S$ that describe a pipeline, which in turn solves task $T$ on dataset $D$. The network also outputs an estimate of pipeline performance $v$. The reinforcement learning algorithm takes the predictions $(P(s, a), v(s))$ produced be the neural network and uses them in the MCTS by running multiple simulations to search for the pipeline sequence $R$ with the best evaluation. An important benefit of this strategy is that it *learns* to synthesize pipelines.

**B**  **Automatic Generation of Task-Based CFG via Meta-Learning**. Manually designed CFGs have many limitations, notably they may not cover all applicable rules and pipeline structures and consequently prevent the search process from exploring desirable pipelines that do not fit the grammar. Furthermore, to create the production rules or patterns in the grammar, a user needs to have knowledge of all the available primitives for a specific task and how they work. For large primitive collections, this is a difficult task, which is compounded for MT-AutoML systems that support multiple problem types. Instead of relying on manually created CFGs, we propose a new strategy that uses meta-learning to derive grammars automatically and on the fly. It does so in two steps: 1) it selects task-specific pipelines and datasets from a meta-learning database (MLDB), and 2) uses these to derive a portfolio of pipeline patterns.

**Selecting Task-Oriented Datasets**. Since AlphaD3M supports different tasks, we need to retrieve from the Marvin MLDB pipelines produced for tasks and datasets similar to the ones we provided as inputs to the AutoML system. For instance, if we want to solve a clustering problem over a dataset $D$, we retrieve the pipelines used for this problem over datasets similar to $D$. To select relevant pipelines for a given problem $P$ over dataset $D$, we use the "task keywords" tag list provided in the problem definition as features that describe the task to be solved, and search Marvin for pipelines that contain a similar set of keywords. The list is encoded as a bag-of-words (BOW). Since the set is small and most of the tags are non-standard words, e.g., *collaborativeFiltering, timeSeries*, it is possible to obtain accurate matches with this simple approach.

Given the set of relevant pipelines $R_P$, we select a subset $R_{PD}$ containing pipelines that were applied on datasets similar to $D$. To determine whether two datasets are similar, we use dataset features including semantic types (e.g., categorical, date-time) and missing values, and encode them using one-hot encoding. Datasets are compared using cosine similarity.

The current implementation uses 16 unique semantic types detected by the datamart_profiler (Datamart Profiler Library, 2021). In contrast to other approaches like TabSim (Habibi et al., 2020), or StruBERT (Trabelsi et al., 2022), AlphaD3M uses semantic types because, in the grammar, it defines components to handle the dataset's features, such as categorical or date-time encoders, and these components are strongly related to semantic types. Also, these approaches focus on tabular datasets, AlphaD3M handles other types of datasets, like image and text datasets. Finally, running these approaches is a very time-consuming task.

**Creating a Portfolio of Patterns**. After identifying similar datasets, the next step is to select the best pipelines to create a portfolio of pipeline patterns. To select these AlphaD3M takes into consideration pipeline performance for different datasets. Some datasets are more challenging than others – the performance of a pipeline can vary widely for different datasets. To properly compare pipeline

performance, AlphaD3M uses a strategy based on the *average distance to minimum* (ADTM) (Wistuba et al., 2015), which transforms the performance to the distance to the best-observed performance scaled between 0 and 1. In contrast to ADTM, which uses the misclassification rate, AlphaD3M uses the actual performance (the score) of the pipelines and thus, it applies the *average distance to maximum* instead to select the best pipelines. It then transforms the primitives within the pipelines to their classes. For instance, the primitive *imputer.SKlearn* belongs to the class *IMPUTATION*. If there is a pipeline with this structure: [*imputer.SKlearn svm.SKlearn*], it is converted to this pattern: [*IMPUTATION CLASSIFICATION*]. Unlike Feurer et al. (2021), which creates a unique portfolio of pipelines in an offline phase, AlphaD3M creates the portfolio online, based on the query task and dataset. Also, the output is a portfolio of patterns, not of static pipelines, which allows more flexibility to construct pipelines. These patterns are used as production rules of the grammar. Algorithm 1 in the Appendix describes the process of building the grammar.

**C  Prioritization of Primitives**. When a data scientist builds an ML pipeline, they start this process using primitives that are known to perform well. For example, XGBoost or Random Forests are good initial candidates for classification tasks. AlphaD3M follows this intuition to identify good candidate primitives for a specific task, using the data from Marvin. This prior knowledge about promising primitives can be helpful to find better pipelines faster.

Similar to Ono et al. (2021), AlphaD3M uses Pearson Correlation (PC) to estimate how much a primitive contributes to the score of the pipeline. However, instead of using the raw scores, it uses the ADTMs values because they are scaled across different datasets. AlphaD3M estimates the primitive importance using PC between the primitive indicator vector $p$ ($p_i = 1$ if pipeline $i$ contains the primitive in question and $p_i = 0$ otherwise) and the pipeline score vector $s$, where $s_i$ is the score for pipeline $i$. Since $p$ and $s$ are dichotomous and quantitative variables, respectively, the Point-Biserial Correlation coefficient (PBC) Sheskin (2003) is an appropriate correlation measure – it is mathematically equivalent to the PC but can be calculated with fewer operations. The correlation values are normalized between 0 and 1 (using min-max normalization).

AlphaD3M calculates these correlations for the primitives at two levels: (a) global, when it considers all the pipelines, and (b) local, when it considers only the pipelines for each pattern. The main goal is to estimate how important a primitive is for all the pipelines and each pattern. Primitives with higher values of importance should have priority during the search of pipelines. Algorithm 2 describes the process of calculating the primitive importance values in detail (see the Appendix). To prioritize the usage of potential primitives in AlphaD3M, it includes these values of primitive importance in the MCTS formula:

$$U(s, a) = Q(s, a) + c(\alpha P(s, a) + (1 - \alpha)R(a)) \frac{\sqrt{N(s)}}{1 + N(s, a)} \tag{1}$$

where $Q(s, a)$ is the expected reward for action $a$ (selection of primitive $a$) from state $s$, $N(s, a)$ is the number of times action $a$ was taken from state $s$, $N(s)$ is the number of times state $s$ was visited. $P(s, a)$ are the probabilities predicted by the neural network over actions $a$ from a state $s$, $c$ is a constant which determines the amount of exploration, $R(a) = G(a) * L(a)$, $G(a)$ and $L(a)$ are the global and local importance of the action $a$, and $\alpha$ is a coefficient to keep the trade-off between $R(a)$ and $P(s, a)$.

**D  Decoupled Hyperparameter Tuning**. Hyperparameter tuning is an essential part of fitting machine learning models (Bergstra et al., 2011; Snoek et al., 2015; Dolatnia et al., 2016). This is also the case for end-to-end ML pipelines that target different tasks, and all primitives contain hyperparameters, not just the estimators.

AlphaD3M performs hyperparameter tuning as an independent task, after the pipelines are constructed. It uses Bayesian optimization, which is the state-of-the-art for hyperparameter tuning

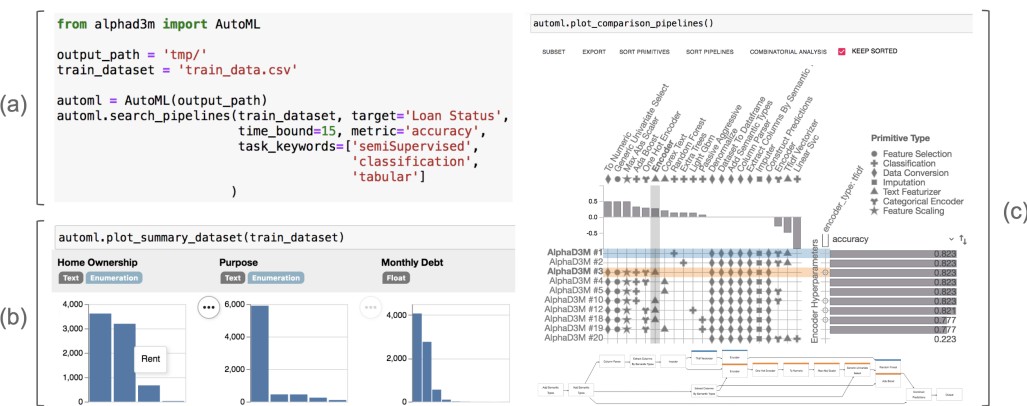

Figure 2: (a) A code snippet to solve a semi-supervised classification task. (b) AlphaD3M allows users to inspect the contents of the input dataset, including column statistics and data types. (c) Analyzing ML pipelines through the integration with PipelineProfiler.

(Bergstra and Bengio, 2012; Snoek et al., 2015; Dolatnia et al., 2016) and was shown to outperform manual setting of parameters, grid search, and random search (Bergstra and Bengio, 2012; Turner et al., 2021).

**Tuning Top-$k$ Pipelines**. AlphaD3M synthesizes and evaluates the pipelines using primitives with default values for hyperparameters. The pipelines are then ranked by performance, and the top-k pipelines are selected for tuning. AlphaD3M uses Sequential Model-Based Algorithm Configuration (SMAC) (Lindauer et al., 2022), a Python library for Bayesian optimization. It approximates a probability model of the performance outcome given a parameter configuration that is updated from a history of executions. AlphaD3M selects the Gaussian Processes models from SMAC to minimize an arbitrary acquisition function using the Expected Improvement criterion to choose the parameter values for each iteration until a condition (number of iterations) is met. The acquisition function is designed to normalize the performance metric used to synthesize the pipelines between zero and one, as the pipeline execution evaluations increase, the acquisition function gets closer to zero. SMAC requires a set of unique parameters to assign values during its tuning procedure. Since AlphaD3M considers multiple primitives with identical names, it constructs an internal hierarchical nomenclature of parameters and designs their dependencies using ConfigSpace.

### 3.3 The API

We have developed a Python-based API that supports the process of building and exploration of ML pipelines within a Jupyter Notebook environment. The API is integrated with the D3M AutoML systems and supports various dataset formats such as raw CSV, D3M, and OpenML. Model synthesis can be done with a few lines of code, as shown in Figure 2(a). The API allows users to (a) define a problem, (b) explore summaries of their input dataset, (c) summarize the produced pipelines and (d) analyze and compare pipelines with respect to their performance scores and prediction outputs. We describe the main components of the API below.

**Problem Definition**. To build a predictive model, AlphaD3M needs a problem specification that describes a prediction problem, specifically: (a) the training dataset; (b) a target variable, i.e., what should be predicted by the predictive model; (c) the maximum running time that controls how long the search can take (to control the use of computational resources); (d) the desired performance metric; and (e) a list of task keywords that specify the kind of prediction task and, therefore, the techniques that should be used to solve the prediction problem. Figure 2(a) shows an example of how to define a problem in AlphaD3M.

Table 2: Comparison of MT-AutoML systems with respect to the number of supported task types, winner pipelines, and average rank by each system.

|  | AlphaD3M | Auto$^n$ML | Ensemble | Aika | Distil | Autoflow | Axolotl | Drori et al. (2019) |
|---|---|---|---|---|---|---|---|---|
| Unique ML tasks supported | 17 | 16 | 15 | 17 | 15 | 16 | 14 | 2 |
| Winner pipelines | 49 | 39 | 30 | 21 | 20 | 11 | 10 | 7 |
| Average rank | 2.85 | 2.89 | 2.90 | 3.99 | 4.68 | 5.32 | 5.73 | 6.85 |

**Data Exploration**. To build good predictive models, it is important to identify data attributes that lead to accurate predictions. The API provides multiple tools for data exploration. For example, it shows different visualizations (compact, detail, and column views) that summarize the content of tabular datasets (see Figure 2 (b)).

**Pipeline Summary**. After the pipeline search is complete, users can display a leaderboard, train individual pipelines with the complete data, perform predictions and evaluate them against a held-out dataset.

**Pipeline Exploration**. Users can analyze the produced pipelines using the PipelineProfiler Ono et al. (2021), which is fully integrated into AlphaD3M as shown in Figure 2(c). PipelineProfiler is a visual analytics tool that enables users to compare and explore the pipelines generated by the AutoML systems.

**Pipeline Refinement and Deployment**. AlphaD3M allows users to save and load pipelines, enabling users to reload them later and perform analyses without having to re-run the AutoML search. They can load the saved pipelines at any time for training or testing purposes. In addition, users can export pipelines to Python code. This gives them more control and the ability to modify (and customize) the automatically generated pipelines (e.g., change hyperparameters, or replace a classifier primitive). More information about the API can be found on the documentation web page: `https://alphad3m.readthedocs.io/en/latest/api.html`.

## 4 Evaluation

To demonstrate the effectiveness of AlphaD3M and its ability to handle a rich set of ML tasks, we compared AlphaD3M with state-of-the-art AutoML systems using two dataset collections. We also present use cases to show how useful, flexible, and easy to use AlphaD3M is.

### 4.1 Comparing AutoML Systems

**D3M Datasets**. This collection contains challenging datasets and cover a wide variety of tasks (a total of 17 task types) and data types (see Table 3). We evaluated all the systems using train and test splits. In most of the cases, the sizes are 0.8 and 0.2 for the train and test splits, respectively (see the dataset's repository [2] for details). For each dataset, we ran the systems over the train split for one hour, a time-bound used by others works (Erickson et al., 2020; Feurer et al., 2021). After that, we evaluated the best pipeline produced by each system in the test split. For this experiment, we used 1 GPU (GeForce GTX 1080 Ti), 14 CPU cores (Intel Xeon E5-2695 v4, 2.10 GHz), and 56 GB memory.

Table 2 shows the number of supported task types (ML tasks), winner pipelines (i.e., pipelines with the best performance for a given dataset), and the average rank by each AutoML system (rank of each system among the 8 AutoML systems applied to each dataset). If two or more systems produce pipelines that tie in the best score, all of them are considered winner pipelines. As we can see, AlphaD3M and Aika were able to solve 17 out of 17 unique tasks, obtaining the best coverage. We also evaluated the effectiveness of AlphaD3M. It had the best overall performance, producing the best pipeline for 49 datasets with the best average rank (2.85). Analyzing the support for each

---

[2]`https://datasets.datadrivendiscovery.org/d3m/datasets`

Table 3: Number of datasets by task type and number of solved datasets by each AutoML system for all task types covered by the D3M datasets.

| ML Task | AlphaD3M | Auto$^n$ML | Ensemble | Aika | Distil | Autoflow | Axolotl | Drori et al. (2019) |
|---|---|---|---|---|---|---|---|---|
| Tabular Classification (20) | 20 | 19 | 18 | 20 | 18 | 17 | 13 | 20 |
| Tabular Regression (11) | 11 | 11 | 11 | 8 | 9 | 6 | 5 | 9 |
| Image Classification (9) | 9 | 8 | 9 | 9 | 7 | 7 | 2 | 0 |
| Image Regression (1) | 1 | 1 | 1 | 1 | 1 | 1 | 1 | 0 |
| Text Classification (9) | 9 | 9 | 9 | 9 | 8 | 8 | 9 | 0 |
| Audio Classification (2) | 2 | 2 | 2 | 2 | 1 | 2 | 2 | 0 |
| Graph Matching (3) | 3 | 3 | 3 | 3 | 2 | 2 | 2 | 0 |
| Time series Forecasting (13) | 13 | 13 | 13 | 13 | 2 | 12 | 10 | 0 |
| Link Prediction (3) | 3 | 3 | 3 | 3 | 2 | 2 | 2 | 0 |
| Collaborative Filtering (1) | 1 | 0 | 1 | 1 | 0 | 1 | 0 | 0 |
| Time series Classification (19) | 19 | 19 | 19 | 17 | 19 | 15 | 19 | 0 |
| Community Detection (3) | 3 | 3 | 0 | 2 | 2 | 1 | 0 | 0 |
| Video Classification (2) | 2 | 2 | 2 | 2 | 0 | 2 | 2 | 0 |
| Vertex Classification (4) | 4 | 4 | 4 | 4 | 4 | 4 | 4 | 0 |
| Object Detection (2) | 2 | 2 | 0 | 1 | 1 | 0 | 0 | 0 |
| Semisupervised Classification (6) | 6 | 6 | 6 | 3 | 6 | 4 | 3 | 0 |
| LUPI (4) | 4 | 4 | 4 | 4 | 4 | 4 | 4 | 0 |

task type individually in Table 3, we can see that AlphaD3M was able to produce valid pipelines for all the datasets and it solved more datasets than the other systems. Even though AlphaD3M is inspired by Drori et al. (2019), in Table Table 2 and Table 3, we can clearly see the difference between them, AlphaD3M handles a larger number of tasks and produces many more winned pipelines. This shows that the different components of AlphaD3M are effective at handling the larger search spaces required by MT-AutoML systems. The detailed scores obtained by each system in all the D3M datasets and the average rank by tasks can be found in Table 4 and Table 5 (Appendix).

Additionally, we calculated the number of winner pipelines for the top-3 systems only in the datasets where all of them produced pipelines. AlphaD3M, Ensemble, and Auto$^n$ML systems got 48, 42, and 38, respectively. These results confirm that the superior performance of AlphaD3M is not solely due to its support for a broader range of ML tasks.

We performed an ablation study to analyze the contribution of each component of AlphaD3M on a random sample of five D3M datasets for classification tasks[2] (datasets for which AlphaD3M obtained the best, average and worst performances). Figure 3 shows the best scores for each dataset reached by the full AlphaD3M and the versions with some components removed (or replaced). As we can see, using all components leads to the best results.

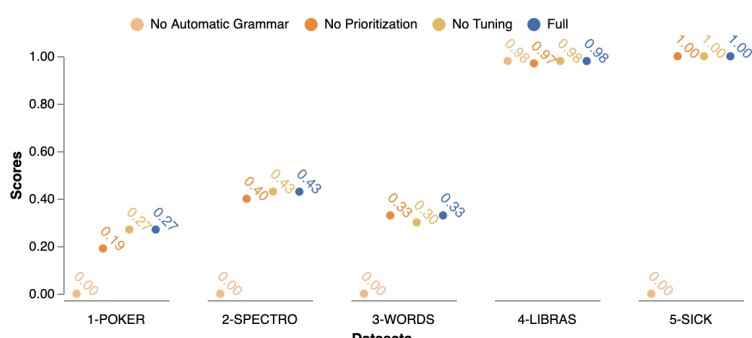

Figure 3: Ablation study for the different components of AlphaD3M.

To evaluate the importance of the automatic grammar, we replaced it with the manually-designed grammar used in Drori et al. (2019). For *POKER*, *SPECTRO*, *WORDS*, and *SICK* datasets, when the manual grammar was used, AlphaD3M was not able to produce valid pipelines, which highlights the importance of automatically generating the grammar. These datasets contain multiple types of features like text, DateTime, etc., which were not covered by the manually-constructed

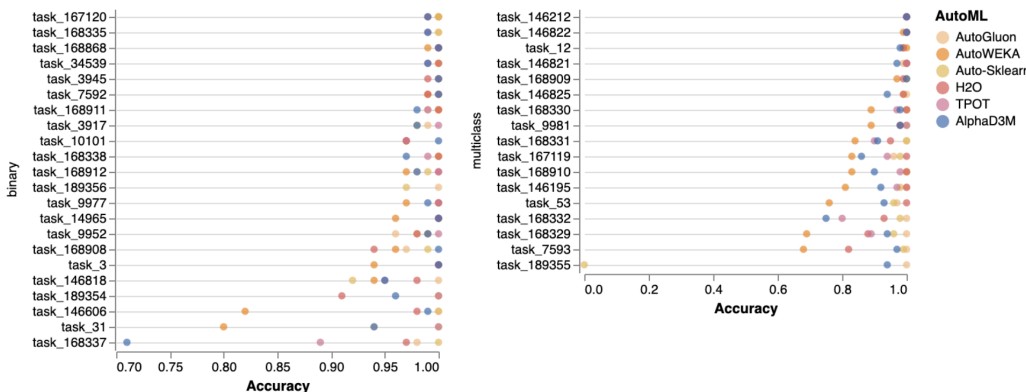

Figure 4: Performance of AutoML systems in OpenML Benchmark. X-axis shows the accuracy values (normalized by the best score), and Y-axis shows the IDs of the OpenML tasks.

grammar. The prioritization of primitives also plays an important role in AlphaD3M. When this feature was not used, the performance decreased, e.g. in *POKER*, *SPECTRO*, and *LIBRAS* datasets. As we can see in Figure 3, in most of the datasets, when we removed the hyperparameter tuning component, AlphaD3M obtained the same results. This suggests that the heuristic used by AlphaD3M (tuning only the top-$k$ pipelines) may miss good pipelines that would attain better performance after tuning. In future work, we plan to investigate alternative strategies for hyperparameter tuning that attain a better balance of computational cost and pipeline performance.

**OpenML Benchmark**. Similar to Erickson et al. (2020), we compared our system with AutoWEKA, TPOT, H2O, AutoGluon, and Auto-Sklearn 2.0 (hereinafter referred to as Auto-Sklearn) on the 39 OpenML datasets (Gijsbers et al., 2019). This corpus contains a variety of datasets intended to represent real-world data science problems and covers binary and multiclass classification tasks. We used AMLB (Gijsbers et al., 2022) to compare the systems, running them locally for one hour using 1 fold split and accuracy as the optimization metric. For this experiment, we used 4 CPU cores (Intel Xeon Platinum 8268 Processor, 2.9 GHz) and 32 GB memory.

Figure 4 shows the scores (normalized by the best score) of all the systems (the detailed scores can be found in Tables 6 and 7 in the Appendix). As we can see, AlphaD3M produced pipelines whose performance is on par with the other AutoML systems. We also calculated the average rank for all the systems for the 39 datasets. AlphaD3M got 3.64 of average rank, while Auto-Sklearn, AutoGluon, H2O, TPOT, and AutoWEKA got 2.08, 2.33, 3.08, 3.72, and 5.10, respectively. To understand better these numbers, we also estimated the performance gain of the pipelines found by AlphaD3M against pipelines generated by other systems. The average gain of AlphaD3M for the OpenML datasets was +0.001, which shows that, in general, AlphaD3M attained good results for this collection. We analyzed the 3 datasets (`task_146195`, `task_167119` and `task_168331`) for which AlphaD3M generated pipelines with performance lower than other systems. This happened because these datasets are imbalanced with multiple classes. The performance of AlphaD3M for these could be improved with the inclusion of primitives to handle imbalanced datasets. This underscores the importance of being able to add primitives to AutoML systems.

Concerning the coverage, it is important to highlight that AlphaD3M succeeded for 38 datasets. Auto-Sklearn, AutoGluon, H2O, TPOT, and AutoWEKA solved 39, 39, 34, 29, and 28 datasets, respectively. As pointed out by Gijsbers et al. (2022), the results of Auto-Sklearn on the OpenML datasets must be considered very carefully, since there could be an overlap between the datasets used in its meta-learning process and the ones used in the evaluation. It's important to highlight that none of the OpenML datasets are included in the version of Marvin that was used by AlphaD3M in these experiments.

## 4.2 Use Cases

**Pivoting across ML tasks**.  Predicting hostile actions against ships and mariners worldwide is important to prevent piracy and prosecute the aggressors. Consider that an analyst from the U.S. National Geospatial-Intelligence Agency (NGA) is building a model using the Anti-Shipping Activity Messages dataset (ASAM, 2021). She wants to identify which records mention guns and which records do not. This is a non-trivial problem since a variety of terms (e.g., pistol, rifle, etc.) indicate whether a gun is present. This dataset contains 8,000 documents, of which 1,400 were annotated. She started by using AlphaD3M to create models using the 1,400 labeled documents setting the model search to 1 hour. AlphaD3M derived high-quality pipelines – the best pipeline had 0.90 of F1. However, she wondered whether these pipelines could be further improved, in particular, by leveraging the 6,600 unlabeled documents through semi-supervised learning. AlphaD3M supports a wide range of tasks, including semi-supervised learning – users just need to add the keyword "semiSupervised" as a parameter. The user then ran a new experiment using the 1,400 labeled and 6,000 unlabeled instances as a training dataset. The results improved from 0.90 to 0.95 of F1. These experiments show that by using AlphaD3M, data scientists can improve the results, pivoting from one task (classification) to another (semi-supervised classification) very quickly.

**Reducing pipeline execution time through models exploration**.  Using content analysis and predictive modeling for conflict assessment is a common approach for conflict analysts to guide policy-making decisions D'Orazio (2020). Consider a conflict analyst trying to categorize explosion events that involve terrorist activities. She uses the explosion events dataset (Raleigh et al., 2010) that contains 20,000 articles describing events that involve terrorist activities. An article is relevant if it describes attacks involving explosions. To create classification models, she ran AlphaD3M for 1 hour. The system synthesized high-quality pipelines, with F1 values around 0.9. To identify the most suitable pipeline, she used the PipelineProfiler to explore the derived models. She observed that the top-10 pipelines had similar scores but their execution times were above 800 seconds. To address this problem, she tried a different strategy: combining progressive sampling and active learning to reduce the number of training data from 20,000 to 3,200 documents. Then, she re-ran AlphaD3M using the smaller set as the training dataset, while keeping the rest of the workflow unchanged. The top F1 score improved from 0.91 to 0.96 and the time from 800 to 125 seconds.

## 5  Conclusions

We introduced AlphaD3M, an MT-AutoML library that automatically synthesizes end-to-end pipelines for 17 ML tasks and 6 different data types. AlphaD3M introduces new methods to automatically derive grammars and prioritize primitives, which are essential for effectively managing the large space MT-AutoML systems must search. In addition, AlphaD3M embraces a user-in-the-loop approach, through an API that allows the users to explore the input data and the derived ML pipelines, as well as customized the pipelines. We presented a detailed experimental evaluation that compares our approach to several state-of-the-art AutoML systems over different problems and datasets. The results suggest that AlphaD3M is effective: not only does it solve a larger number of problem types, but it also derives pipelines with performance that is superior or on par with those derived by other systems.

Although AlphaD3M's approach is primitive-agnostic, so far, it only relies on the D3M primitives to build ML pipelines. We plan to extend AlphaD3M by including additional state-of-the-art and more-recent primitives, e.g., models published in HuggingFace or PyTorch Hub repositories. Moreover, we would like to improve the system interoperability with existing open-source primitives that use standard APIs such as the well-known scikit-learn's `fit-predict` API.

**Acknowledgements**. This work was partially supported by the DARPA D3M program.  Any opinions, findings, conclusions, or recommendations expressed in this material are those of the authors and do not necessarily reflect the views of DARPA.

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

## A  Broader Impact Statement

AlphaD3M can potentially strengthen the efforts in democratizing data science by broadening the application of automated predictive pipelines. Subject experts can create their own pipelines and explore them in the context of an ethical framework. Its interoperable software infrastructure enables external auditing and improves the trust and interpretability of synthesized pipelines. The search space management mechanism also allows efficient resource allocation and helps to prototype pipelines before performing high energy-consuming model training.


## C  Additional Details

### C.1  Algorithms

Algorithm 1 describes the process of building the grammar. $getVectorTK$ and $getVectorST$ represent the BOW and one-hot encoding functions, respectively. The best values empirically calculated for the thresholds $t_{sim}$ and $t_{perf}$ are 0.8 and 0.5, respectively.

---
**Algorithm 1** Grammar Builder
---

**Input:** Marvin datasets $D$, query dataset $q$, threshold $t$
Initialize $S = []$ // Similar datasets
**for** $d_i$ **in** $D$ **do**
    $simTK = cosineSimilarity(getVectorTK(d_i), getVectorTK(q))$
    **if** $simTK > t_{sim}$ **then**
        $simST = cosineSimilarity(getVectorST(d_i), getVectorST(q))$
        **if** $simST > t_{sim}$ **then**
            Add $d_i$ to $S$
Initialize $P = calculateADTM(S)$
Initialize $R = []$ // Production Rules
**for** $p_i$ **in** $P$ **do**
    **if** $performance(p_i) > t_{perf}$ **then**
        $r_i = convertToPattern(p_i))$
        Add $r_i$ to $R$
**return** $R$

---

Algorithm 2 describes the process of calculating the primitive importance values in detail. For instance, the primitive importance values calculated for XGBoost and Random Forrest are 0.62 and 0.56, whereas for Nearest Centroid and K-Nearest Neighbors the values are 0.46 and 0.44. It shows that the importance values can be used as an indicator to prioritize the usage of primitives.

---
**Algorithm 2** Primitives Importance
---

**Input:** Pipelines $P$, Patterns $T$
Initialize $R = getPrimitives(P)$
Initialize $G, L = []$ // Global and Local correlations
**for** $r_i$ **in** $R$ **do**
    $pc = PearsonCorrelation(r_i, P)$
    $npc = normalize(pc)$
    Add $npc$ to $G$
**for** $t_i$ **in** $T$ **do**
    $p_i = getPipelines(t_i, P)$
    $R = getPrimitives(t_i, p_i)$
    **for** $r_i$ **in** $R$ **do**
        $pc = PearsonCorrelation(r_i, R)$
        $npc = normalize(pc)$
        Add $npc$ to $L$
**return** $(G, L)$

---

## C.2 Grammars

Different tasks require different grammars. For instance, the algorithms needed to solve time-series and semi-supervised classification problems have a different structure and use a different set of primitives. Consequently, specialized grammars and production rules are needed for each task. Manually creating these grammars is time-consuming and error-prone, and relying on these grammars can limit the effectiveness of the AutoML systems with respect to problem coverage and quality of the derived pipelines.

Figure 5 shows an excerpt of a grammar automatically generated in AlphaD3M to solve classification problems. The start symbol (*S*) is the starting point from which all the production rules can be derived. In the grammar, the terminal *'primitive'* can be any of the available algorithms in AlphaD3M, and *'E'* represents the empty symbol.

```
S ::= CATEGORICAL_ENCODER TEXT_FEATURIZER DATA_CONVERSION IMPUTATION CLASSIFICATION
S ::= TEXT_FEATURIZER CATEGORICAL_ENCODER FEATURE_SCALING IMPUTATION FEATURE_SELECTION CLASSIFICATION
S ::= IMPUTATION TEXT_FEATURIZER CATEGORICAL_ENCODER FEATURE_SCALING FEATURE_SELECTION CLASSIFICATION
S ::= IMPUTATION TEXT_FEATURIZER CATEGORICAL_ENCODER DIMENSIONALITY_REDUCTION CLASSIFICATION
S ::= DATA_STRUCTURE_ALIGNMENT IMPUTATION CLASSIFICATION
S ::= IMPUTATION FEATURE_SCALING CLASSIFICATION
S ::= IMPUTATION FEATURE_SELECTION CLASSIFICATION
S ::= IMPUTATION DIMENSIONALITY_REDUCTION CLASSIFICATION
IMPUTATION ::= 'primitive' | 'E'
CATEGORICAL_ENCODER ::= 'primitive' | 'E'
FEATURE_SCALING ::= 'primitive' | 'E'
FEATURE_SELECTION ::= 'primitive' | 'E'
DIMENSIONALITY_REDUCTION ::= 'primitive' | 'E'
DATA_CONVERSION ::= 'primitive'
TEXT_FEATURIZER ::= 'primitive'
DATA_STRUCTURE_ALIGNMENT ::= 'primitive'
CLASSIFICATION ::= 'primitive'
```

Figure 5: Excerpt of a grammar automatically generated by AlphaD3M for classification tasks

In Figure 6, you can see the manual grammar used in the experiments. This grammar was proposed by Drori et al. (2019). To generate this grammar for classification and regression tabular tasks, a developer was asked to review manually the primitives to group them into categories. For instance, the primitives *decision_tree.SKlearn* and *random_forest.SKlearn* were grouped into the category 'CLASSIFICATION'. Then, using his knowledge in ML, he created the production rules of the grammar using these categories.

```
S ::= CLASSIFICATION_TASK | REGRESSION_TASK
CLASSIFICATION_TASK ::= CLASSIFICATION | DATA_CLEANING CLASSIFICATION | DATA_TRANSFORMATION CLASSIFICATION |
                        DATA_CLEANING DATA_TRANSFORMATION CLASSIFICATION
REGRESSION_TASK ::= REGRESSION | DATA_CLEANING REGRESSION | DATA_TRANSFORMATION REGRESSION |
                    DATA_CLEANING DATA_TRANSFORMATION REGRESSION
CLASSIFICATION ::= 'primitive'
REGRESSION ::= 'primitive'
DATA_CLEANING ::= 'primitive' DATA_CLEANING | 'E'
DATA_TRANSFORMATION ::= 'primitive' DATA_TRANSFORMATION | 'E'
```

Figure 6: Manual Grammar

## C.3 Experiments

In Table 4, we can see the scores obtained by all AutoML systems developed in the D3M program, including a majority voting ensemble system, on a collection of 112 datasets[2]. This collection

contains challenging datasets that go beyond the simple tabular data and cover a wide variety of tasks and data types.

Table 4: Scores obtained by AlphaD3M and the other AutoML systems developed in the D3M program.

| Dataset | AlphaD3M | Auto$^n$ML | Ensemble | Aika | Distil | Autoflow | Axolotl | Drori |
|---|---|---|---|---|---|---|---|---|
| 124_120_mnist_8747 | 0.98 | 0.94 | 0.46 | 0.18 | 0.94 | 0.11 | - | - |
| 124_138_cifar100_1858 | 0.67 | 0.48 | 0.42 | 0.12 | 0.48 | 0.01 | - | - |
| 124_16_fashion_mnist | 0.90 | 0.83 | 0.84 | 0.12 | 0.85 | 0.10 | - | - |
| 124_174_cifar10_MIN | 0.88 | 0.82 | 0.84 | 0.27 | 0.80 | 0.10 | - | - |
| 124_188_usps_MIN | 0.96 | 0.95 | 0.94 | 0.26 | 0.92 | 0.18 | 0.11 | - |
| 124_214_coil20_MIN | 0.99 | 0.99 | 0.99 | 0.85 | 0.97 | - | - | - |
| 124_95_uc_merced_land_use_MIN | 0.90 | - | 0.72 | 0.52 | - | 0.05 | 0.33 | - |
| 1491_one_hundred_plants_margin_MIN | 0.80 | 0.79 | 0.88 | 0.92 | 0.75 | 0.83 | 0.81 | 0.83 |
| 1567_poker_hand_MIN | 0.90 | 0.84 | 0.28 | 0.48 | 0.12 | 0.13 | - | 0.27 |
| 185_baseball_MIN | 0.66 | 0.70 | 0.65 | 0.68 | 0.68 | 0.67 | 0.66 | 0.64 |
| 196_autoMpg_MIN | 6.57 | 9.12 | 5.74 | 11.95 | 7.49 | 6.01 | 15.36 | 7.03 |
| 22_handgeometry_MIN | 0.24 | 0.23 | 0.23 | 0.14 | 0.80 | 0.36 | 0.36 | - |
| 26_radon_seed_MIN | 0.02 | 0.02 | 0.24 | 0.03 | 0.02 | 0.06 | 1.40 | 0.02 |
| 27_wordLevels_MIN | 0.32 | 0.28 | 0.28 | 0.32 | 0.29 | 0.27 | 0.26 | 0.27 |
| 299_libras_move_MIN | 0.98 | - | - | 0.48 | - | - | 0.98 | 0.97 |
| 30_personae_MIN | 0.62 | 0.65 | 0.65 | 0.62 | 0.61 | 0.55 | 0.61 | - |
| 313_spectrometer_MIN | 0.43 | 0.37 | 0.37 | 0.30 | 0.32 | 0.33 | 0.23 | 0.40 |
| 31_urbansound_MIN | 0.93 | 0.93 | 0.91 | 0.75 | 0.92 | 0.77 | 0.49 | - |
| 32_fma_MIN | 0.55 | 0.57 | 0.34 | 0.28 | - | 0.11 | 0.11 | - |
| 32_wikiqa_MIN | 0.00 | 0.02 | 0.14 | 0.13 | 0.50 | - | 0.13 | - |
| 38_sick_MIN | 1.00 | 1.00 | - | 1.00 | - | - | 0.49 | 1.00 |
| 4550_MiceProtein_MIN | 1.00 | 1.00 | 1.00 | 0.99 | 1.00 | 1.00 | 1.00 | 1.00 |
| 49_facebook_MIN | 0.88 | 0.87 | 0.87 | 0.87 | 0.87 | 0.88 | 0.44 | - |
| 534_cps_85_wages_MIN | 20.11 | 20.35 | 22.07 | 23.15 | 24.86 | 21.44 | - | 20.70 |
| 56_sunspots_MIN | 34.55 | 11.82 | 8.64 | 8.45 | 58.30 | 9.40 | 90.60 | - |
| 56_sunspots_monthly_MIN | 64.61 | 41.18 | 46.86 | 41.04 | - | 62.20 | 27.74 | - |
| 57_hypothyroid_MIN | 0.96 | 0.98 | 0.99 | 0.98 | 0.74 | 0.99 | 0.97 | 0.98 |
| 59_LP_karate_MIN | 0.93 | 0.45 | 0.83 | 0.83 | 0.45 | 0.45 | 0.93 | - |
| 59_umls_MIN | 0.92 | 0.94 | 0.94 | 0.94 | 0.94 | 0.70 | 0.73 | - |
| 60_jester_MIN | 4.25 | - | 4.24 | 4.15 | - | 4.51 | - | - |
| 66_chlorineConcentration_MIN | 0.82 | 0.86 | 0.81 | 0.52 | 0.78 | 0.68 | 0.23 | - |
| 6_70_com_amazon_MIN | 0.85 | 0.85 | - | 0.85 | 0.85 | - | - | - |
| 6_86_com_DBLP_MIN | 0.72 | 0.72 | - | 0.72 | 0.72 | - | - | - |
| JIDO_SOHR_Articles_1061 | 0.98 | 0.94 | 0.94 | 0.81 | 0.56 | 0.60 | 0.64 | - |
| JIDO_SOHR_Tab_Articles_8569 | 1.00 | 0.99 | 1.00 | 1.00 | 0.56 | 1.00 | 1.00 | - |
| LL0_1100_popularkids_MIN | 0.42 | 0.45 | 0.38 | 0.38 | 0.40 | 0.44 | - | 0.47 |
| LL0_186_braziltourism_MIN | 0.14 | 0.35 | 0.36 | 0.17 | 0.24 | 0.20 | 0.34 | 0.16 |
| LL0_207_autoPrice_MIN | $4.89 \cdot 10^6$ | $5.76 \cdot 10^6$ | $6.04 \cdot 10^6$ | $3.76 \cdot 10^7$ | $5.36 \cdot 10^6$ | $5.43 \cdot 10^6$ | $1.56 \cdot 10^8$ | $5.81 \cdot 10^6$ |
| LL0_acled_reduced_MIN | 0.83 | 0.88 | 0.89 | 0.84 | 0.91 | 0.85 | 0.74 | 0.91 |
| LL0_jido_reduced_MIN | 0.90 | 0.89 | 0.91 | 0.90 | 0.90 | 0.90 | - | 0.90 |
| LL1_2734_CLIR | 0.88 | 0.50 | 0.52 | 0.88 | - | - | 0.50 | - |
| LL1_336_MS_Geolife_transport_MIN | 0.60 | 1.00 | 0.99 | - | 0.85 | - | 0.98 | - |
| LL1_336_MS_Geolife_transport_separate | 0.67 | 1.00 | 0.99 | - | 0.86 | - | 0.99 | - |
| LL1_3476_HMDB_actio_recognition_MIN | 0.11 | 1.00 | 0.90 | 0.11 | - | 0.48 | 0.08 | - |
| LL1_50words_MIN | 0.35 | 0.55 | 0.56 | 0.41 | 0.51 | 0.45 | 0.35 | - |
| LL1_726_TIDY_GPS_carpool | 0.54 | 0.58 | 0.58 | 0.46 | 0.59 | - | 0.63 | - |
| LL1_736_population_spawn_MIN | 1636.12 | 1806.40 | 1804.76 | 1644.26 | - | 2845.89 | - | - |
| LL1_736_population_spawn_simpler_MIN | 1346.10 | 1490.15 | 3669.54 | 1347.65 | 1323.72 | 1550.40 | 19887.20 | - |
| LL1_736_stock_market_MIN | 7.64 | 1.49 | 8.69 | 1.75 | - | 30.66 | - | - |
| LL1_ACLED_TOR_online_behavior_MIN | 0.40 | 0.05 | 0.44 | 0.64 | 0.43 | 0.66 | 0.08 | 0.40 |
| LL1_Adiac_MIN | 0.75 | 0.70 | 0.73 | 0.54 | 0.67 | 0.70 | 0.49 | - |
| LL1_ArrowHead_MIN | 0.75 | 0.82 | 0.78 | 0.72 | 0.65 | 0.55 | 0.72 | - |
| LL1_CONFLICT_3457_atrocity | 9.53 | 6.75 | 11.43 | 12.84 | - | 17.21 | 13.91 | - |
| LL1_Cricket_Y_MIN | 0.52 | 0.54 | 0.59 | 0.52 | 0.62 | 0.53 | 0.45 | - |
| LL1_DIC28_net_MIN | 0.84 | 0.80 | 0.80 | 0.80 | 0.80 | 0.84 | - | - |
| LL1_ECG200_MIN | 0.90 | 0.87 | 0.87 | 0.86 | 0.91 | 0.85 | 0.86 | - |
| LL1_EDGELIST_net_nomination_MIN | 0.99 | 0.66 | 0.85 | 0.94 | 0.66 | 0.35 | 0.84 | - |
| LL1_ElectricDevices_MIN | 0.54 | 0.42 | 0.46 | 0.06 | 0.44 | 0.27 | 0.31 | - |
| LL1_FISH_MIN | 0.80 | 0.87 | 0.89 | 0.73 | 0.84 | 0.86 | 0.78 | - |
| LL1_FaceFour_MIN | 0.84 | 0.83 | 0.71 | 0.55 | 0.65 | 0.40 | 0.66 | - |

| Dataset | AlphaD3M | $\text{Auto}^n\text{ML}$ | Ensemble | Aika | Distil | Autoflow | Axolotl | Drori |
|---|---|---|---|---|---|---|---|---|
| LL1_GS_process_classification_tab_MIN | 0.80 | 0.80 | 0.80 | 0.80 | 0.80 | 0.73 | - | 0.81 |
| LL1_GS_process_classification_text_MIN | 0.65 | 0.80 | 0.65 | 0.80 | 0.80 | 0.76 | 0.80 | - |
| LL1_GT_actor_group_association_MIN | 0.25 | 0.13 | 0.17 | 0.13 | - | - | - | - |
| LL1_HandOutlines_MIN | 0.89 | 0.91 | 0.90 | 0.88 | 0.88 | 0.88 | 0.88 | - |
| LL1_Haptics_MIN | 0.43 | 0.42 | 0.44 | 0.42 | 0.41 | 0.45 | 0.42 | - |
| LL1_ItalyPowerDemand_MIN | 0.93 | 0.95 | 0.95 | 0.95 | 0.95 | 0.91 | 0.90 | - |
| LL1_MIL_MUSK | 0.68 | 0.77 | 0.83 | 0.67 | 0.80 | 0.80 | - | 0.72 |
| LL1_MIL_Mutagenesis | 0.80 | 0.73 | 0.72 | 0.71 | 0.70 | 0.63 | - | 0.79 |
| LL1_MITLL_synthetic_vora_E_2538 | 0.29 | 0.53 | 0.52 | 0.50 | 0.31 | 0.44 | - | 0.38 |
| LL1_Meat_MIN | 0.95 | 0.94 | 0.88 | 0.92 | 0.88 | 0.17 | 0.95 | - |
| LL1_OSULeaf_MIN | 0.53 | 0.44 | 0.52 | 0.77 | 0.45 | 0.47 | 0.32 | - |
| LL1_PHEM_Monthly_Malnutrition_MIN | 10.63 | 9.56 | 9.39 | 9.73 | - | 12.18 | - | - |
| LL1_PHEM_weekly_malnutrition_MIN | 3.34 | 4.32 | 3.45 | 2.94 | - | 4.23 | 4.18 | - |
| LL1_TXT_CLS_3746_newsgroup_MIN | 0.60 | 0.46 | 0.55 | 0.48 | 0.60 | 0.45 | 0.23 | - |
| LL1_TXT_CLS_SST_Binary | 0.73 | 0.82 | 0.82 | 0.55 | - | 0.51 | 0.53 | - |
| LL1_TXT_CLS_airline_opinion_MIN | 0.81 | 0.80 | 0.81 | 0.80 | 0.81 | 0.72 | 0.72 | - |
| LL1_TXT_CLS_apple_products_sent_MIN | 0.73 | 0.71 | 0.72 | 0.72 | 0.73 | 0.66 | 0.69 | - |
| LL1_VID_UCF11_MIN | 0.99 | 0.99 | 0.25 | 0.27 | - | 0.02 | 0.08 | - |
| LL1_VTXC_1343_cora_MIN | 0.61 | 0.04 | 0.22 | 0.17 | 0.04 | 0.13 | 0.52 | - |
| LL1_VTXC_1369_synthetic_MIN | 0.95 | 0.22 | 0.33 | 0.21 | 0.22 | 0.19 | 0.48 | - |
| LL1_ViEWS_CM_S1 | 0.69 | 1.20 | 0.90 | 0.72 | 0.75 | 2.52 | - | 0.82 |
| LL1_ViEWS_PGM_S1 | 0.02 | 0.04 | 0.02 | - | 0.02 | 0.02 | 0.30 | 0.02 |
| LL1_bigearth_landuse_detection | 0.90 | 0.96 | 0.76 | 0.65 | 0.21 | - | - | - |
| LL1_bn_fly_drosophila_medulla_net_MIN | 0.24 | 0.24 | - | - | - | 0.19 | - | - |
| LL1_h1b_visa_apps_7480 | 0.44 | 0.47 | 0.43 | 0.44 | 0.41 | 0.41 | 0.47 | 0.42 |
| LL1_net_nomination_seed_MIN | 0.99 | 0.99 | 0.96 | 0.94 | 0.99 | 0.34 | 0.46 | - |
| LL1_penn_fudan_pedestrian_MIN | 0.94 | 0.94 | - | 0.94 | 0.94 | - | - | - |
| LL1_retail_sales_total_MIN | 1989.19 | 1921.54 | 1941.06 | 1966.30 | 1992.17 | - | 1971.76 | 2022.41 |
| LL1_terra_canopy_height_s4_100_MIN | 113.04 | 68.44 | 39.02 | 52.21 | - | 79.86 | 343.27 | - |
| LL1_terra_canopy_height_s4_70_MIN | 104.92 | 547.94 | 126.06 | 136.32 | - | 169.63 | 136.98 | - |
| LL1_terra_canopy_height_s4_80_MIN | 112.95 | 92.95 | 32.57 | 74.59 | - | 111.49 | 74.54 | - |
| LL1_terra_canopy_height_s4_90_MIN | 117.13 | 85.73 | 35.12 | 60.44 | - | 104.49 | 60.45 | - |
| LL1_terra_leaf_angle_mean_s4_MIN | 0.04 | 0.09 | 0.05 | 0.04 | - | - | 0.05 | - |
| LL1_tidy_terra_panicle_detection_MIN | 0.01 | 0.03 | - | - | - | - | - | - |
| SEMI_1040_sylva_prior_MIN | 0.93 | 0.90 | 0.93 | - | 0.92 | - | - | - |
| SEMI_1044_eye_movements_MIN | 0.52 | 0.57 | 0.61 | 0.55 | 0.60 | 0.53 | 0.54 | - |
| SEMI_1053_jm1_MIN | 0.26 | 1.00 | 0.16 | - | 0.16 | 0.41 | - | - |
| SEMI_1217_click_prediction_small_MIN | 0.04 | 0.03 | 0.04 | - | 0.17 | - | - | - |
| SEMI_1459_artificial_characters_MIN | 0.68 | 0.99 | 0.83 | 0.99 | 0.67 | 0.61 | 0.52 | - |
| SEMI_155_pokerhand_MIN | 0.58 | 0.66 | 0.60 | 0.05 | 0.64 | 0.50 | 0.51 | - |
| kaggle_music_hackathon_MIN | 21.88 | 17.56 | 19.64 | 24.24 | 21.79 | - | - | 21.85 |
| loan_status_MIN | 0.40 | 0.50 | 0.51 | 0.44 | 0.33 | - | 0.48 | 0.46 |
| political_instability_MIN | 0.81 | 0.89 | 0.89 | 0.89 | 0.89 | - | 0.88 | - |
| uu1_datasmash_MIN | 1.00 | 1.00 | 1.00 | 1.00 | 0.61 | 1.00 | 1.00 | - |
| uu2_gp_hyperparameter_estimation_MIN | 0.89 | 0.88 | 0.57 | 0.89 | - | - | - | 0.89 |
| uu3_world_development_indicators_MIN | $2.39 \cdot 10^{10}$ | $5.54 \cdot 10^{12}$ | $4.12 \cdot 10^{12}$ | - | $4.40 \cdot 10^{12}$ | - | - | - |
| uu3_world_development_indicators_raw | $7.83 \cdot 10^{13}$ | $1.04 \cdot 10^{12}$ | $5.22 \cdot 10^{11}$ | - | - | - | - | - |
| uu4_SPECT_MIN | 0.00 | 0.92 | 0.92 | 0.90 | 0.89 | 0.90 | 0.78 | - |
| uu5_heartstatlog_MIN | 0.70 | 0.69 | 0.72 | 0.62 | 0.61 | 0.72 | 0.67 | - |
| uu6_hepatitis_MIN | 0.00 | 0.47 | 0.89 | 0.40 | 0.27 | 0.31 | 0.44 | - |
| uu7_pima_diabetes_MIN | 0.59 | 0.57 | 0.60 | 0.57 | 0.60 | 0.63 | 0.57 | - |
| uu_101_object_categories_MIN | 0.95 | 0.89 | 0.84 | 0.34 | - | 0.10 | - | - |

The average rank values obtained by different AutoML systems for each task type in the D3M datasets can be seen in Table 5. These datasets contain a total of 17 unique ML tasks.

Table 5: Average rank values by task obtained by different AutoML systems.

| Task | AlphaD3M | Auto$^n$ML | Ensemble | Aika | Distil | Autoflow | Axolotl | Drori |
|---|---|---|---|---|---|---|---|---|
| Image Classification | 1.11 | 2.78 | 2.78 | 4.56 | 4.33 | 6.22 | 7.44 | 8.00 |
| Tabular Classification | 3.75 | 3.30 | 3.35 | 3.85 | 4.85 | 4.65 | 5.85 | 3.55 |
| Tabular Regression | 2.27 | 3.18 | 3.00 | 5.73 | 4.27 | 5.73 | 7.54 | 4.36 |
| Image Regression | 4.00 | 2.00 | 2.00 | 1.00 | 7.00 | 5.00 | 5.00 | 8.00 |
| Text Classification | 2.56 | 3.33 | 2.22 | 3.00 | 3.56 | 5.78 | 4.33 | 8.00 |
| Audio Classification | 1.50 | 1.00 | 3.50 | 5.00 | 5.50 | 5.00 | 6.00 | 8.00 |
| Graph Matching | 1.00 | 3.33 | 3.00 | 2.33 | 4.67 | 3.33 | 6.33 | 8.00 |
| Time series Forecasting | 3.38 | 3.62 | 2.62 | 2.23 | 7.31 | 5.08 | 5.08 | 8.00 |
| Link Prediction | 3.33 | 2.33 | 2.33 | 1.67 | 4.67 | 6.67 | 5.00 | 8.00 |
| Collaborative Filtering | 3.00 | 8.00 | 2.00 | 1.00 | 8.00 | 4.00 | 8.00 | 8.00 |
| Time series Classification | 3.26 | 2.26 | 2.16 | 4.68 | 3.79 | 5.32 | 4.53 | 8.00 |
| Community Detection | 1.00 | 1.00 | 8.00 | 3.33 | 3.33 | 6.33 | 8.00 | 8.00 |
| Video Classification | 2.50 | 1.00 | 3.00 | 3.50 | 8.00 | 4.50 | 5.50 | 8.00 |
| Vertex Classification | 1.00 | 4.00 | 3.25 | 4.25 | 4.00 | 6.50 | 3.50 | 8.00 |
| Object Detection | 1.50 | 1.00 | 8.00 | 4.50 | 4.50 | 8.00 | 8.00 | 8.00 |
| Semisupervised Classification | 3.50 | 2.33 | 2.33 | 6.00 | 2.83 | 6.00 | 6.83 | 8.00 |
| LUPI | 5.25 | 3.00 | 1.25 | 4.50 | 5.00 | 2.50 | 4.75 | 8.00 |

Table 6 and Table 7 show the raw and normalized scores (normalized by the best score) obtained by each system on the 39 datasets of the OpenML AutoML Benchmark (Gijsbers et al., 2019). This benchmark represents real-world data science problems and covers binary and multiclass classification tasks. Additionally, Table 6 shows the gain of AlphaD3M regarding the other systems.

Table 6: Raw scores obtained by AlphaD3M and the other AutoML systems.

| Dataset | AutoGluon | AutoWEKA | Auto-Sklearn | H2O | TPOT | AlphaD3M | Gain |
|---|---|---|---|---|---|---|---|
| task_10101 | 0.76 | 0.76 | 0.76 | 0.76 | 0.76 | 0.79 | 0.03 |
| task_12 | 0.98 | 0.98 | 0.98 | 0.98 | - | 0.96 | -0.01 |
| task_146195 | 0.88 | 0.71 | 0.86 | 0.88 | 0.85 | 0.81 | -0.03 |
| task_146212 | 1.00 | 1.00 | 1.00 | 1.00 | 1.00 | 1.00 | 0.00 |
| task_146606 | 0.74 | 0.60 | 0.73 | 0.72 | - | 0.73 | 0.03 |
| task_146818 | 0.91 | 0.86 | 0.84 | 0.90 | 0.87 | 0.87 | -0.01 |
| task_146821 | 0.99 | 1.00 | 1.00 | 1.00 | 1.00 | 0.97 | -0.03 |
| task_146822 | 0.97 | 0.97 | 0.97 | 0.97 | 0.98 | 0.97 | 0.00 |
| task_146825 | 0.91 | - | 0.91 | 0.90 | - | 0.86 | -0.05 |
| task_14965 | 0.91 | 0.88 | 0.91 | 0.91 | 0.91 | 0.91 | 0.00 |
| task_167119 | 0.92 | 0.80 | 0.94 | 0.96 | 0.90 | 0.83 | -0.08 |
| task_167120 | 0.51 | 0.51 | 0.51 | 0.51 | - | 0.51 | -0.00 |
| task_168329 | 0.40 | 0.27 | 0.38 | 0.35 | 0.35 | 0.37 | 0.02 |
| task_168330 | 0.73 | 0.65 | 0.73 | 0.73 | 0.70 | 0.72 | 0.01 |
| task_168331 | 0.73 | 0.62 | 0.73 | 0.69 | 0.66 | 0.66 | -0.02 |
| task_168332 | 0.56 | - | 0.54 | 0.51 | 0.44 | 0.41 | -0.10 |
| task_168335 | 0.94 | - | 0.94 | - | 0.93 | 0.94 | -0.00 |
| task_168337 | 0.84 | - | 0.86 | 0.83 | 0.77 | 0.61 | -0.21 |
| task_168338 | 1.00 | - | 1.00 | 1.00 | 0.99 | 0.97 | -0.03 |
| task_168868 | 0.99 | 0.99 | 0.99 | 1.00 | 0.99 | 0.99 | 0.00 |
| task_168908 | 0.74 | 0.73 | 0.76 | 0.72 | - | 0.77 | 0.03 |
| task_168909 | 0.99 | 0.96 | 0.99 | 0.98 | - | 0.99 | 0.01 |
| task_168910 | 0.72 | 0.60 | 0.72 | 0.72 | 0.71 | 0.65 | -0.04 |
| task_168911 | 0.81 | 0.82 | 0.82 | 0.82 | 0.81 | 0.81 | -0.01 |
| task_168912 | 0.93 | 0.92 | 0.95 | 0.95 | 0.95 | 0.94 | -0.00 |
| task_189354 | 0.67 | - | 0.67 | 0.61 | 0.67 | 0.65 | -0.01 |
| task_189355 | 0.94 | - | 0.00 | - | - | 0.88 | 0.41 |
| task_189356 | 0.71 | - | 0.69 | - | - | - | - |
| task_3 | 0.99 | 0.93 | 0.99 | 1.00 | 0.99 | 0.99 | 0.01 |
| task_31 | 0.77 | 0.66 | 0.82 | - | 0.82 | 0.77 | 0.00 |
| task_34539 | 0.95 | - | 0.95 | 0.95 | 0.95 | 0.95 | -0.01 |
| task_3917 | 0.87 | - | 0.86 | - | 0.88 | 0.86 | -0.01 |
| task_3945 | 0.98 | - | 0.98 | 0.98 | 0.98 | 0.98 | 0.00 |
| task_53 | 0.86 | 0.67 | 0.85 | 0.88 | - | 0.82 | 0.01 |
| task_7592 | 0.87 | 0.87 | 0.87 | 0.86 | 0.87 | 0.87 | 0.00 |
| task_7593 | 0.97 | 0.66 | 0.96 | 0.80 | - | 0.95 | 0.10 |
| task_9952 | 0.88 | 0.91 | 0.90 | 0.90 | 0.91 | 0.91 | 0.01 |
| task_9977 | 0.98 | 0.95 | 0.97 | 0.98 | 0.97 | 0.96 | -0.00 |
| task_9981 | 0.94 | 0.86 | 0.96 | 0.94 | 0.96 | 0.94 | 0.01 |

Table 7: Normalized scores obtained by AlphaD3M and the other AutoML systems.

| Dataset | AutoGluon | AutoWEKA | Auto-Sklearn | H2O | TPOT | AlphaD3M |
|---|---|---|---|---|---|---|
| task_10101 | 0.97 | 0.97 | 0.97 | 0.97 | 0.97 | 1.00 |
| task_12 | 0.99 | 1.00 | 0.99 | 0.99 | - | 0.98 |
| task_146195 | 1.00 | 0.81 | 0.98 | 1.00 | 0.97 | 0.92 |
| task_146212 | 1.00 | 1.00 | 1.00 | 1.00 | 1.00 | 1.00 |
| task_146606 | 1.00 | 0.82 | 1.00 | 0.98 | - | 0.99 |
| task_146818 | 1.00 | 0.94 | 0.92 | 0.98 | 0.95 | 0.95 |
| task_146821 | 0.99 | 1.00 | 1.00 | 1.00 | 1.00 | 0.97 |
| task_146822 | 1.00 | 0.99 | 1.00 | 1.00 | 1.00 | 1.00 |
| task_146825 | 1.00 | - | 0.99 | 0.99 | - | 0.94 |
| task_14965 | 1.00 | 0.96 | 1.00 | 1.00 | 1.00 | 1.00 |
| task_167119 | 0.96 | 0.83 | 0.98 | 1.00 | 0.94 | 0.86 |
| task_167120 | 1.00 | 1.00 | 1.00 | 0.99 | - | 0.99 |
| task_168329 | 1.00 | 0.69 | 0.96 | 0.88 | 0.89 | 0.94 |
| task_168330 | 1.00 | 0.89 | 1.00 | 1.00 | 0.97 | 0.98 |
| task_168331 | 1.00 | 0.84 | 1.00 | 0.95 | 0.90 | 0.91 |
| task_168332 | 1.00 | - | 0.98 | 0.93 | 0.80 | 0.75 |
| task_168335 | 1.00 | - | 1.00 | - | 0.99 | 0.99 |
| task_168337 | 0.98 | - | 1.00 | 0.97 | 0.89 | 0.71 |
| task_168338 | 1.00 | - | 1.00 | 1.00 | 0.99 | 0.97 |
| task_168868 | 1.00 | 0.99 | 1.00 | 1.00 | 1.00 | 1.00 |
| task_168908 | 0.97 | 0.96 | 0.99 | 0.94 | - | 1.00 |
| task_168909 | 1.00 | 0.97 | 1.00 | 0.99 | - | 1.00 |
| task_168910 | 1.00 | 0.83 | 1.00 | 1.00 | 0.98 | 0.90 |
| task_168911 | 0.99 | 1.00 | 1.00 | 1.00 | 0.99 | 0.98 |
| task_168912 | 0.98 | 0.97 | 0.99 | 1.00 | 1.00 | 0.98 |
| task_189354 | 1.00 | - | 1.00 | 0.91 | 1.00 | 0.96 |
| task_189355 | 1.00 | - | 0.00 | - | - | 0.94 |
| task_189356 | 1.00 | - | 0.97 | - | - | - |
| task_3 | 1.00 | 0.94 | 1.00 | 1.00 | 1.00 | 1.00 |
| task_31 | 0.94 | 0.80 | 1.00 | - | 1.00 | 0.94 |
| task_34539 | 1.00 | - | 1.00 | 1.00 | 0.99 | 0.99 |
| task_3917 | 0.99 | - | 0.98 | - | 1.00 | 0.98 |
| task_3945 | 1.00 | - | 1.00 | 0.99 | 1.00 | 1.00 |
| task_53 | 0.97 | 0.76 | 0.96 | 1.00 | - | 0.93 |
| task_7592 | 1.00 | 0.99 | 1.00 | 0.99 | 1.00 | 1.00 |
| task_7593 | 1.00 | 0.68 | 0.99 | 0.82 | - | 0.97 |
| task_9952 | 0.96 | 0.99 | 0.98 | 0.98 | 1.00 | 0.99 |
| task_9977 | 1.00 | 0.97 | 1.00 | 1.00 | 1.00 | 0.99 |
| task_9981 | 0.98 | 0.89 | 1.00 | 0.98 | 1.00 | 0.98 |

