# OpenReview forum: "AlphaD3M: An Open-Source AutoML Library for Multiple ML Tasks"
_automl.cc/AutoML/2023/ABCD_Track — AutoML 2023 (ABCD Track)_

### Official Review · Reviewer_9nZi · 2023-05-04

**Potential Impact On The Field Of Automl Rating:** 3
**Technical Quality And Correctness Rating:** 3
**Clarity Rating:** 3

**Summary Of Contributions:**

The paper introduces AlphaD3M, an AutoML library that automatically synthesizes end-to-end pipelines for 17 different machine learning tasks and six data types. The main contributions include new methods for automatically deriving grammars and prioritizing primitives, which help manage the search space effectively. The system also adopts a user-in-the-loop approach with an API that allows users to explore input data and derived machine learning pipelines. The authors perform a comprehensive experimental evaluation comparing AlphaD3M to various state-of-the-art AutoML systems over different problems and datasets. The results demonstrate that AlphaD3M is effective in solving a larger number of problem types and generating pipelines with performance that is either superior or on par with those derived by other systems.

**Actions Required To Increase Overall Recommendation:**

Addressing questions under overall review section - please see weaknesses, additional questions.

**Clarity:**

Overall the clarity of the paper is good. The paper is well-written and clearly presents the contributions and methodology sections. The authors also provide detailed information on the experiments, including the datasets, comparison systems, and evaluation metrics.

There are a few areas where clarity could be further improved:
1. The paper discusses the user-in-the-loop approach and the API for exploring input data and derived ML pipelines, but more details or examples on the actual user interactions would have been helpful to illustrate how users can effectively utilize AlphaD3M.


**Overall Review:**

Strengths:
1. AlphaD3M addresses a wide range of ML tasks and data types, which is a significant advantage over many existing AutoML systems that only support a limited number of tasks.
2. The automatic construction of grammars and prioritization of primitives are novel and help in handling larger search spaces.
3. Provides a thorough experimental evaluation using two dataset collections (D3M Datasets and OpenML Benchmark) and compares the performance of AlphaD3M with state-of-the-art AutoML systems.
4. Inclusion of real use cases helps demonstrate the practical utility of AlphaD3M in real-world scenarios and its ease of use.
5. The user-in-the-loop approach, supported by an API for exploring input data and derived ML pipelines, enhances usability and makes it a more practical solution for data scientists.

Weaknesses:
1. The current version of AlphaD3M relies only on the D3M primitives to build ML pipelines. Incorporating additional state-of-the-art and more recent primitives from other repositories would be beneficial in improving the system's capabilities.
2. The paper does not provide an in-depth analysis of the scalability of the system. It is unclear how the system would perform with even larger datasets or more complex problems.
3. Integration with other libraries or tools: The paper does not discuss potential integration with other state-of-the-art and more recent primitives, such as models published in HuggingFace or PyTorch Hub repositories. Including such primitives would extend the capabilities of AlphaD3M and make it more appealing to a broader audience.
4. More details on below two items would add value:
4.1 Automatic grammar construction:
a. How does the automatic grammar construction algorithm deal with complex data types and situations where relationships between features are not easily discernible?
b. How do you ensure the quality of the generated grammar, especially when dealing with noisy or incomplete data? Are there any quality metrics or validation methods applied during the grammar construction process?
4.2 Prioritization of primitives:
a. How does the system learn or update the prioritization of primitives over time, especially when new primitives are introduced or existing primitives are updated?
b. Can you elaborate on the criteria used to prioritize primitives? How are these criteria determined, and are they adaptable to different domains or problem types?
c. How does the system handle cases where multiple primitives have similar performance or impact on the pipeline? How does it determine which primitives to prioritize in such cases?

Additional questions for the authors:
1. How does AlphaD3M handle cases where the dataset has missing values or noisy data? Are there any preprocessing steps or data cleaning techniques incorporated into the system?
2. How does the system perform when handling imbalanced datasets? Are there any specific techniques applied to address this issue?
3. Can you provide more details on the execution times of AlphaD3M compared to other AutoML systems, especially when dealing with larger datasets?
4. Are there any plans to extend AlphaD3M to support additional ML tasks such as regression, time series forecasting, or reinforcement learning?

**Potential Impact On The Field Of Automl:**

I believe there could be a significant potential impact on the field of AutoML as it presents an innovative approach to synthesizing end-to-end pipelines for a wide range of machine learning tasks and data types. AlphaD3M's ability to automatically derive grammars and prioritize primitives addresses the challenge of effectively managing the search space, a critical aspect in AutoML general usecases. The user-in-the-loop approach provides an additional layer of practicality and usability, which could inspire future AutoML systems. I'm certain that researchers and practitioners in the AutoML domain are likely to cite this paper for its contributions, as it not only demonstrates a competitive performance compared to existing state-of-the-art AutoML systems but also showcases the flexibility and ease of use that AlphaD3M offers.

**Review Confidence:**

4: You are confident in your assessment, but not absolutely certain. It is unlikely, but not impossible, that you did not understand some parts of the submission or that you are unfamiliar with some pieces of related work.

**Review Rating:**

7: Weak Accept: Technically sound paper with moderate-to-high impact, with perhaps some minor flaws.

**Review Summary:**

The paper presents AlphaD3M, an AutoML library designed to synthesize end-to-end pipelines for a wide range of machine learning tasks and data types. The system introduces new methods for automatically deriving grammars and prioritizing primitives, improving its ability to handle larger search spaces. The paper provides a detailed experimental evaluation comparing AlphaD3M with other state-of-the-art AutoML systems, demonstrating its superior performance and coverage.

However, the paper could benefit from discussing potential integration with other state-of-the-art libraries and tools could expand the capabilities of AlphaD3M and make it more appealing to a broader audience. Also adding more in-depth explanations of some components, such as automatic grammar construction and prioritization of primitives would strengthen the paper.

**Technical Quality And Correctness:**

Few minor limitations:
1. Authors compare their system to the results computed by the AutoGluon-Tabular project, which include other vendors like H20.ai, Auto-WEKA, auto-sklearn, TPOT, AutoGluon etc. Although the comparison is reasonable, it would have been more comprehensive to run AlphaD3M directly against the other AutoML systems to ensure fair comparison conditions.
2. The ablation study is limited to a random sample of five D3M datasets for classification tasks. A larger sample size or more diverse dataset types could have provided a more robust assessment of the system's components.

---

### Official Review · Reviewer_MRQE · 2023-05-06

**Potential Impact On The Field Of Automl Rating:** 4
**Technical Quality And Correctness Rating:** 4
**Clarity Rating:** 3
**Actions Required To Increase Overall Recommendation:** 1. Include Auto-Sklearn 2.0 in the co…

**Summary Of Contributions:**

This paper introduces a new AutoML library that can handle a wider range of machine learning problems and data types compared to existing libraries. It's designed to be user-friendly, even for novices, thanks to the automatic derivation of CFG through meta-learning. Additionally, to enhance the quality of the generated pipelines, the authors include a notion of prioritization for primitives, building on previous research.

**Clarity:**

Overall, the paper is easy to read and understand. However, a few minor changes could enhance its clarity:
* Figure 3 lacks an explanation of the score used; e.g., Figure 4 makes it evident that the score is the accuracy. Moreover, it would be helpful to cite the datasets used in Figure 3 (e.g., BASEBALL, LIBRAS, etc.) to make it easier for readers to identify them.
* Table 2 should include the references to the evaluated systems, even if they do not have associated published papers.
* It's unclear why Auto-Sklearn 2.0 was not evaluated in Figure 4 despite being part of the references.

**Overall Review:**

The proposed library has the potential to facilitate further advancements in AutoML systems due to several key advantages. Firstly, all the components of the library are clearly defined and justified, making it easier to extend and develop. Secondly, it supports a broader range of ML tasks and datatypes compared to other existing systems. Additionally, the performance of the pipelines generated by the library is either comparable to or better than other systems. Furthermore, the GitLab repository provides enough examples of how to use the library for various types of problems, and the API is well-documented. Lastly, the main components of the library have been ablated to demonstrate their contribution.

**Potential Impact On The Field Of Automl:**

While many of the techniques employed in the proposed library have been used in previous works, the overall modifications and combinations of these techniques enable the library to handle a wider range of problems while maintaining its ease of use. Additionally, each component, such as automatic CFG generation, primitive prioritization, pipeline generation, and hyperparameter tuning, is clearly defined and justified, making it easy to extend the library.

**Reproducibility (Optional):**

The library is open-sourced on GitLab, and the authors provide numerous examples of how to use it, and comprehensive API documentation. However, the authors do not include the necessary information to replicate the experiments discussed in the paper but they mention this explicitly in the "Submission Checklist."

**Review Confidence:**

4: You are confident in your assessment, but not absolutely certain. It is unlikely, but not impossible, that you did not understand some parts of the submission or that you are unfamiliar with some pieces of related work.

**Review Rating:**

8: Accept: Technically sound paper with major impact, with perhaps some minor flaws.

**Review Summary:**

Overall, the advantages of the proposed library are clear, the analysis is comprehensive, and the paper is well-written and easy to understand. Furthermore, the paper provides scope for further improvements and extensions to the proposed library, indicating its potential for future advancements in AutoML systems.

**Technical Quality And Correctness:**

The paper does not present any technical issues. It provides examples of its use cases and comprehensive API documentation. The authors' experiments demonstrate the library's superiority over existing ones. Additionally, the paper includes ablation studies of the primary components, namely automatic CFG generation and primitive prioritization, which provide a better understanding of their significance.

---

### Official Review · Reviewer_9Utf · 2023-05-10

**Potential Impact On The Field Of Automl Rating:** 3
**Technical Quality And Correctness Rating:** 2
**Clarity Rating:** 3

**Summary Of Contributions:**

The authors present AlphaD3M, an open source Python based AutoML system that can handle a wide variety of task types with only a few lines of code. AlphaD3M incorporates reinforcement learning & meta-learning for pipeline selection trained on Marvin, a meta-corpus of 600 datasets and 2.5M pipelines. The meta-learning is informed based on task keywords which are manually declared by the user as part of the fit call. AlphaD3M splits hyperparameter tuning into a post-pipeline selection stage and leveraging existing bayesian optimization techniques to tune the final result. The authors emphasize AlphaD3M's application within user-in-the-loop scenarios, and present two benchmarks that showcase AlphaD3M's performance compared to existing AutoML systems, claiming comparable or superior performance to state-of-the-art AutoML systems.

**Actions Required To Increase Overall Recommendation:**

The AlphaD3M system is compelling and novel, but the paper falters in the experiments section. I am open to increasing my score substantially if the experiments are given more clear definitions, exact result numbers are provided in appendix tables, and average ranks are calculated to support the performance claims of the system. I would be even more open to increasing my score if code was provided to be able to reproduce and verify the experimental results. Finally, discussion should be added on how overfitting was avoided in case any benchmark datasets were used for meta-learning.

**Clarity:**

In general the paper is easy to follow and is well written. However, critical information is frequently missing in the experimental setup and results discussion, and the appendix does not provide sufficient additional information to make up for these gaps.

- API-wise, `search_pipelines` and `train` seem to be oddly separated. `train` lacks any time-limit bounds. At what point does AlphaD3M produce a valid solution that the user can predict with? How is time limit respected if `train` fit time is being ignored?
- Why task keywords not automatically identified? Why have to manually specify? Is it realistic users would know what to specify? This seems like a barrier to ease of use.
- On the website: Lacking examples/tutorials on some task types, and lacking description of how to format data for them.

## Task support for video/image/text/audio

There is little if any description of how to format data for these types of tasks nor the models used. These fields have advanced substantially in recent years, so it is unclear if the results achievable with AlphaD3M are meaningfully competitive with out-of-the-box pre-trained models + fine-tuning. Since there isn't discussion on this topic in the paper beyond mentioning future extensions to add HuggingFace models, it is hard to quantify how useful in practice the claimed support for these modalities are at present, and comparisons with other D3M reliant systems aren't sufficient to verify relevance compared to simple fine-tuning baselines and open-source systems currently leveraging these pre-trained models such as AutoGluon and (recently) FLAML. Benchmarking all of these different task types with modern alternatives would certainly be challenging, so I can understand why it wasn't conducted, but this may be useful to include as potential future work.

## Reducing pipeline execution time through models exploration.
- It is unclear how this demonstrates strengths of AlphaD3M. Couldn't you do this with any AutoML system that is able to produce a solution to the task? To me this reads as "User made the data better, fed to AutoML, achieved better result".

## Semisupervised Learning
Semisupervised support is a great feature to have, however its implementation is not covered in detail, and only anecdotal evidence is provided that is enhances model quality beyond pure supervised learning. While I would like additional information here, it is honestly a topic worthy of an entire paper in and of itself, so I understand why it was only talked about in brief.

**Overall Review:**

This is a solid paper discussing a novel and feature rich open source AutoML system that currently falls short on experimental rigor, reproducibility, and analysis, leading it to make unsupported claims on performance. In particular, it does not mention the risks or mitigation strategies of benchmarking on datasets that may have been used in meta-learning, and does not provide sufficient details to reproduce experiments nor code to do so. Refer to the above comments for details.

**Potential Impact On The Field Of Automl:**

The paper provides a sophisticated AutoML system capable in a few lines of code of handling a wide range of task types and leverages meta-learning and pipeline construction techniques that are useful to the AutoML community. I also very much like the inclusion of semi-supervised learning support, which has the potential to greatly enhance the utility of the system in real-world scenarios.

However the impact is lessened until the benchmark results are analyzed with methods such as average rank and code is provided to reproduce the results. Further, there is little if any discussion on what types of datasets or problems that AlphaD3M performs poorly in and why.

**Reproducibility (Optional):**

I have concerns about reproducibility, specifically that no code was provided to run the experiments and no result files or tables were provided to verify the result numbers.

**Review Confidence:**

5: You are absolutely certain about your assessment. You are very familiar with the related work and checked all the details carefully.

**Review Rating:**

6: Borderline Leaning Accept: Technically sound submission where reasons to accept outweigh reasons to reject. Please use sparingly.

**Review Summary:**

I am inclined to weakly reject the paper in its current form due to the lack of experimental result breakdown and the unsupported claims of "performance that is superior or on par with those derived by other systems." via metrics such as average rank or relative error, along with the lack of code to reproduce the results of any of the experiments.

[Update post-rebuttal] I have increased my score to a 6 (borderline leaning accept) based on the author's responses. The score is contingent on the authors updating the requested tables and figures to support their performance claims in the camera-ready version.

**Technical Quality And Correctness:**

I am generally convinced of the overall AlphaD3M design and implementation. The reasoning provided for the decisions are sound and result in a compelling and novel AutoML system that is relatively easy to use and covers a wide range of tasks.

My primary concerns with the paper stem from the experiments, specifically how they were conducted, their lack of code to reproduce, and how the results were analyzed.

- How to ensure benchmark results are genuine given the meta-learning? Have you checked if the benchmark datasets were used in meta-learning? If so, how does this impact the results and are they still meaningful?

## Table 2
- Table 2 lacks average rank analysis or statistical significance tests. Simply having the most "winning pipelines" is not enough to declare a system as the strongest, as it penalizes systems that are similar to each-other disproportionately (even if they all perform very well, they would split their wins against each-other, making it possible for a weak system to have the most wins). Average rank would be a better indicator as it does not suffer from this problem. Further, it is not discussed how ties are handled in the results table.
- Table 2 lacks analysis on runtime.
- Lacks hardware usage discussion
- Table 2 lacks analysis on where AlphaD3M performs poorly and why.

## OpenML Benchmark:
- "Similarly, we run AlphaD3M for one hour". With what hardware? This is not specified. The original AutoGluon-Tabular paper used `m5.2xlarge` instances with 8 vCPU cores, which is also the default instance used in AutoMLBenchmark.
- Did you run on only 1 fold or all 10 folds? This is important information that was not provided.
- With what evaluation metric?
- Why not present results in a similar format as the original AutoGluon-Tabular paper and include average rank?
- Figure 4: This figure is quite unhelpful. There is no table of exact scores and very similar colors are used for the different AutoML systems, all stacked on top of each-other. Looking solely at this figure, I am unable to extract any meaningful insight.
- No average rank / most wins / etc. summary of results, and thus there is no support to the claim of "AlphaD3M produced pipelines whose performance is on par with the other AutoML systems".
- No training time analysis to verify that the systems are respecting the time limit.
- The figures showcase "Accuracy", but the AutoML systems were not ran to optimize "Accuracy", but rather "ROC_AUC" for binary, and "log_loss" for multiclass in the original AutoGluon-Tabular paper. Thus, it would not be fair to compare on "Accuracy" as the models would be trained differently by the AutoML systems had they known "Accuracy" was the evaluation metric.
- You may want to consider including analysis of similar form to those done in pages 20-27 in https://arxiv.org/pdf/2207.12560.pdf
  - The above paper also includes updated results on AMLB for major AutoML systems, which may be more relevant to compare to than the 2020 results from the AutoGluon-Tabular paper.
  - Why not contribute AlphaD3M to AutoMLBenchmark so it is part of future benchmark papers using AMLB? This would significantly increase the impact and trustworthiness of the system.

## Ablation

- Ablation study of removing all task keywords is missing, this seems like it would be very important, as this appears to be a manual task that a user would have to specify, and thus not all users would know how to specify the keywords properly.
- Why ablation study of removed components covers only 5 datasets? This is not enough to produce meaningful take-aways. Also, would be good to include statistical tests and average rank.

---

### Official Review · Reviewer_ATiu · 2023-05-11

**Potential Impact On The Field Of Automl Rating:** 3
**Technical Quality And Correctness Rating:** 3
**Clarity Rating:** 3

**Summary Of Contributions:**

The paper proposes an open-source Python library, called AlphaD3M, which is able to automatically construct pipelines for 17 different unique learning tasks by combining deep reinforcement learning and meta-learning.

The pipelines found by the AlphaD3M have been compared with other pipelines obtained by other AutoML software/libraries and comparable results have been presented. AlphaD3M is able to find solutions for 17 different learning tasks, while some other AutoML libraries are specialized for a subset of tasks. There are also other libraries such as Aika that work for all included learning tasks.

**Actions Required To Increase Overall Recommendation:**

- Estimate what is the performance gain of pipelines found by AlphaD3M against pipelines generated by other tools.

- Address the minor comments presented above.

**Clarity:**

The paper is well-written and really well-structured. It is easy to follow it.
I have some points that can improve the clarity:

1) Can you please provide arguments why you decide the library to be built based on D3M primitives (maybe because it is related to some of your previous work)?

2) Searching for similar datasets has been performed by one-hot encoding. Can you elaborate more on how this affects the end results if other dataset meta-representation are selected? I believe that this is important for the pipeline. Have you tried other representations as Dataset2vec etc.?

3) The part for prioritization of primitives has been difficult to follow on the first read.






**Overall Review:**

Positive: The paper proposes an open-source Python library, called AlphaD3M, which is able to automatically construct pipelines for 17 different unique learning tasks by combining deep reinforcement learning and meta-learning. The pipelines found by the AlphaD3M have been compared with other pipelines obtained by other AutoML software/libraries and comparable results have been presented. It is really nice written paper. I believe this is a nice contribution to the AutoML community.


Negative: Even benchmarking results are presented on D3M datasets and OpenML benchmark, maybe a statistical analysis should be performed to see the benefit of using the library against the others already available.

**Potential Impact On The Field Of Automl:**

The paper is really well-presented and structured. From the usage of this library, a lot of communities that are doing ML can benefit (especially data scientists who are not aware of hyperparameter tuning). I definitely will try the usage of the library for some application scenarios to experiment with the results.

**Reproducibility (Optional):**

I have not checked the documentation.

**Review Confidence:**

4: You are confident in your assessment, but not absolutely certain. It is unlikely, but not impossible, that you did not understand some parts of the submission or that you are unfamiliar with some pieces of related work.

**Review Rating:**

7: Weak Accept: Technically sound paper with moderate-to-high impact, with perhaps some minor flaws.

**Review Summary:**

The paper is well written, with minor flows that can be explained. The part that is missing is what is the contribution of the library compared to the already existing one. The state-of-the-art is nicely summarized in a table, however, in the benchmarking, it seems that other libraries/tools are able to solve 17 or 16 learning tasks. Maybe you can estimate what is the performance gain in cases when AlphaD3M is superior to pipelines find by other tools.

**Technical Quality And Correctness:**

The paper is well-written including almost all details. The results of automatically finding a good pipeline have been explained for two use cases and also benchmarked on D3M datasets and OpenML benchmark. More explanations about the evaluation, train and test splits (even if you specified in the checklist that is available in references) it is nice to be explained.

Is it fair for only the top-k pipelines to be used for hyper-parameter tuning? Other pipelines with tuned hyper-parameters probably will change the top-k ranked. Have you performed a kind of sensitivity analysis to estimate the loss of doing the selection by using the default parameters?

---

### Official Review · Reviewer_MytT · 2023-05-11

**Potential Impact On The Field Of Automl Rating:** 4
**Technical Quality And Correctness Rating:** 3
**Clarity Rating:** 4
**Actions Required To Increase Overall Recommendation:** See my reviews on technical section f…

**Summary Of Contributions:**

The paper introduces AlphaD3M, a Python library that's capable of automatically constructing machine learning pipelines on a wide range of data types and learning tasks. It works on the D3M primitives, which is compatible with the entire D3M ecosystem. AlphaD3M comes with a flexible API that's accessible to both basic and expert users. It also includes a novel system to generate tasked-based CFG for each specific task, which was shown to outperform manually curated grammar in the experiment section.

**Clarity:**

Overall, I find the paper clear and easy to understand.

Minor improvements:
- On page 3: for the first paragraph in Section 3, it might be a good idea to consider re-order the description of pipelines to match the order in Figure 1
- On page 8: consider converting Figure 3 to a table instead

**Overall Review:**

Pros:
- AlphaD3M is a nicely documented, open-sourced, library with accessible API
- It supports a wide-range of tasks and data types
- It comes with a novel task-based grammar generation system and avoid biases from manually-define grammar, and the author performs ablation study to prove its effectiveness
- It is compatible with the D3M ecosystem


Cons:
- AlphaD3M doesn't show a great performance improvement over existing AutoML libraries
- The script to reproduce benchmark result is missing

**Potential Impact On The Field Of Automl:**

Comparing to the existing AutoML toolkit that mostly focuses on tabular data, AlphaD3M supports a wider types of data, including image, text, audio, etc. It is also able to cover a wider range of tasks. The novel CFG generator remove the need to manually define a grammar for pipeline search, which makes the library more accessible and performant. The use of D3M primitives further ensure the extensibility of the library. If we can further validate the performance of the library, this AlphaD3M could be a one-stop shop for various AutoML needs.

**Reproducibility (Optional):**

The author did not provide the code to re-generate the benchmark.

**Review Confidence:**

3: You are fairly confident in your assessment. It is possible that you did not understand some parts of the submission or that you are unfamiliar with some pieces of related work.

**Review Rating:**

8: Accept: Technically sound paper with major impact, with perhaps some minor flaws.

**Review Summary:**

Overall, AlphaD3M has a potential to be a one-stop AutoML toolkit for general needs because of its capability to support a wide range of tasks.

**Technical Quality And Correctness:**

The author has open-sourced AlphaD3M on GitLab (https://gitlab.com/ViDA-NYU/d3m/alphad3m) and published the Python package on PyPI (which already has 15k downloads).

However, here're some clarifying questions that concerns me on the evaluation section:
1. What's the machine/computational resource being used to run the benchmark?
2. What does "winner pipeline" means in Table 2?
3. For some of the tasks in Figure 4, there's a nontrivial difference between the performance of AlphaD3M and the best approach. Do we have any insights on what caused the difference (or if there's a pattern on the type of datasets that AlphaD3M perform well on)?